# Methyl Jasmonate- and Salicylic Acid-Induced Transcription Factor *ZjWRKY18* Regulates Triterpenoid Accumulation and Salt Stress Tolerance in Jujube

**DOI:** 10.3390/ijms24043899

**Published:** 2023-02-15

**Authors:** Cuiping Wen, Zhong Zhang, Qianqian Shi, Xiaoshan Duan, Jiangtao Du, Cuiyun Wu, Xingang Li

**Affiliations:** 1College of Forestry, Northwest Agriculture and Forestry University, Xianyang 712100, China; 2Research Center for Jujube Engineering and Technology of National Forestry and Grassland Administration, Northwest Agriculture and Forestry University, Xianyang 712100, China; 3Agricultural Genomics Institute at Shenzhen, Chinese Academy of Agricultural Sciences, Shenzhen 518116, China; 4College of Horticulture and Forestry, Tarim University, Alar 843300, China

**Keywords:** triterpenoid biosynthesis, jujube (*Ziziphus jujuba* Mill.), *ZjWRKY18*, silencing, overexpression, salt tolerance

## Abstract

Triterpenoids are important, pharmacologically active substances in jujube (*Ziziphus jujuba* Mill.), and play an important role in the plant’s resistance to abiotic stress. However, regulation of their biosynthesis, and the underlying mechanism of their balance with stress resistance, remain poorly understood. In this study, we screened and functionally characterized the *ZjWRKY18* transcription factor, which is associated with triterpenoid accumulation. The transcription factor is induced by methyl jasmonate and salicylic acid, and its activity was observed by gene overexpression and silencing experiments, combined with analyses of transcripts and metabolites. *ZjWRKY18* gene silencing decreased the transcription of triterpenoid synthesis pathway genes and the corresponding triterpenoid content. Overexpression of the gene promoted the biosynthesis of jujube triterpenoids, as well as triterpenoids in tobacco and *Arabidopsis thaliana*. In addition, *ZjWRKY18* binds to W-box sequences to activate promoters of 3-hydroxy-3-methyl glutaryl coenzyme A reductase and farnesyl pyrophosphate synthase, suggesting that *ZjWRKY18* positively regulates the triterpenoid synthesis pathway. Overexpression of *ZjWRKY18* also increased tolerance to salt stress in tobacco and *Arabidopsis thaliana*. These results highlight the potential use of *ZjWRKY18* to improve triterpenoid biosynthesis and salt stress tolerance in plants, and provide a strong basis for metabolic engineering to improve the content of triterpenoids and breeding of jujube varieties that are resistant to stress.

## 1. Introduction

Triterpenoids are ubiquitous secondary metabolites with various bioactivities, and are essential components of signal transduction [1,2]. Most identified terpenoids are synthesized by plants and are induced by hormones such as methyl jasmonate (MeJA) and salicylic acid (SA) [3,4]. MeJA treatment can promote the accumulation of triterpenoids [5]. Triterpenoids play important roles in plant respiration, photosynthesis, growth and development, and regulation. Additionally, triterpenoids accumulate as macromolecules to resist biotic and abiotic stresses in plants [6]. Because of their special and beneficial biological properties to human health, triterpenoids have wide applications in pharmaceuticals, foods, cosmetics, and industrial biotechnology [7].

Jujube (*Ziziphus jujuba* Mill.) is a medicinal and economical fruit crop in the family *Rhamnaceae*, and is rich in a variety of active substances, including triterpenoids, flavonoids, and alkaloids. Jujube is mainly cultivated on land with high salinity and drought conditions in northern China, and this plant exhibits high tolerance to drought and infertility [8]. A variety of triterpenoids are found in jujube fruits, seeds, and roots, mainly tetracyclic and pentacyclic triterpenoids, and the triterpenoids in jujube fruits have been reported to have anti-inflammatory, anti-cancer, and sleep-improving effects [9]. To date, more than 40 triterpenoids have been isolated from jujube, including ursolic acid and oleanolic acid, which have been reported to reduce blood sugar and promote resistance to oxidation [9]. Although there has been extensive research on the pharmacological activities of jujube, there is little known about the triterpenoid biosynthesis pathways. Triterpenoids are formed via the cytosolic mevalonate (MVA) pathway, which provides isopentenyl diphosphate (IPP) and dimethylallyl diphosphate (DMAPP) via 3-hydroxy-3-methyl glutaryl coenzyme A reductase (HMGR), to synthesize triterpenoid intermediates [10]. IPP and DMAPP form triterpene precursors with the 30-carbon atom squalene via the farnesyl pyrophosphate synthase (FPS) and squalene synthase (SQS) enzymes, and then squalene epoxidase (SQE) and oxidosqualene cyclases (OSCs) act to form pentacyclic triterpenoids [11]. The current understanding of the biosynthetic genes for triterpenoids remains incomplete. Transcription factors (TFs) can regulate the transcription of multiple biosynthetic genes to effect the synthesis and accumulation of terpenoids both at the whole plant level and in culture systems. Regulation by a single TF could affect the expression of several coordinately regulated biosynthetic genes, leading to increased or decreased metabolite production [12,13]. TFs such as WRKYs, MYBs, and bHLHs have been reported to regulate triterpenoid biosynthesis in *P. ginseng*, birch, and *Glycyrrhiza uralensis*, respectively [14,15,16].

WRKY TFs have received much attention for their regulation of stress resistance in a variety of plants. There is increasing evidence that some WRKY TFs regulate the production of secondary metabolites (terpenoids, alkaloids, and phenylpropanoids) by regulating genes involved in the biosynthesis pathways [17,18]. Preliminary transcriptome and metabolome data [19] suggested that WRKY TFs may be involved in triterpenoid regulation. WRKY TFs can be induced by methyl jasmonate (MeJA) and other hormones to promote the synthesis of secondary metabolites [20]. For example, WRKYs regulate terpenoid accumulation in tomato [21], diterpenes momilactone A in rice [22], and polyterpenes in rubber tree [23]. For triterpenoids, *PqWRKY1* regulates triterpenoid pathway genes in *Panax quinquefolia*, including FPS, SQS, and SQE [24]; *PgWRKY4X* positively regulates ginsenoside biosynthesis by activating SQE transcription in *Panax ginseng* [16]. Abiotic stresses can also induce WRKYs to improve stress tolerance in plants [25]. For example, *DgWRKY1* or *DgWRKY3* from *Dendranthema grandiflorum* can enhance salt tolerance in tobacco [26], and overexpression of *GhWRKY17* increased plant tolerance to drought and salt stress [27].

Previous work identified WRKY TFs responding to stress in jujube [28]. However, the ability of WRKY TFs to regulate triterpenoid synthesis in jujube has not been studied. Here, we identified TF *ZjWRKY18* in jujube and described its functional characteristics. The results showed the important role of this transcription factor in triterpenoid synthesis and stress resistance regulation. Further, we demonstrate that *ZjWRKY18* positively regulates triterpenoid biosynthesis and salt tolerance in model plants. Our results suggest that *ZjWRKY18* regulates triterpenoid accumulation via transcriptional modulation of the triterpenoid synthesis pathway and also enhances salt tolerance in plants.

## 2. Results

### 2.1. Identification and Expression Analysis of ZjWRKY18

We use part of the transcriptome and metabolome data from previous research by our group (Zhang et al., 2022) [19]. The selected data is listed in Appendix A. Here, the correlation between WRKY transcription factors and triterpenoid metabolism was analyzed (Appendix A), and ten WRKY transcription factors with high correlations with triterpenoid content were selected (Appendix A). Both MeJA and salicylic acid (SA) treatments were associated with triterpenoid accumulation (Figure 1A). Compared with the non-induced control (0 h), MeJA treatment caused the gradually increased accumulation of triterpenoids, with a maximum at 24 h (approximately 2-fold), and salicylic acid treatment caused a gradual increase, reaching a maximum at 12 h (approximately 3-fold), followed by a gradual decrease at 24 h. Of the two plant hormones, SA induced a higher accumulation of triterpenes than MeJA (Figure 1A). After treatment with MeJA and SA, the expression of *ZjWRKY18* exhibited the highest correlation with the content of triterpenoids, so *ZjWRKY18* was selected for additional screening (Appendix A). We analyzed the *ZjWRKY18* for potential SA and MeJA promoter response elements (Appendix A). We also analyzed *ZjWRKY18* expression under MeJA and SA induction, and observed triterpenoid accumulation. The transcription levels of *ZjWRKY18* in the roots of wild jujube seedlings were induced by MeJA and SA at different times (Figure 1B). Compared with the control, MeJA treatment resulted in an increase in the *ZjWRKY18* transcription level, with a maximum at 24 h, followed by a small decrease in the expression level. However, under SA treatment, the expression of *ZjWRKY18* gradually increased, reached a peak at 12 h, decreased slightly at 24 h, and then showed a slow decline (Figure 1B). Further expression analyses of tissues and fruits showed that the *ZjWRKY18* expression level was highest in the roots, followed by the leaves and fruits, and lowest in stem tissues (Appendix A). Triterpenoid accumulation was induced by both MeJA and SA, and the expression of *ZjWRKY18* was consistent with triterpenoid accumulation in response to induction by MeJA and SA, indicating that *ZjWRKY18* plays an important role in triterpenoid accumulation. 

### 2.2. Phylogenetic and Subcellular Localization Analysis

Sequence analysis revealed that the length of the ORF region of *ZjWRKY18* was 960 bp, encoding 319 amino acids and containing a putative WRKY domain (WRKYGQK). Phylogenetic analysis of *ZjWRKY1*8 showed a high similarity to Group III WRKYs of *Gossypium arboretum* (GaWRKY1), *A. thaliana* (AtWRKY40 and AtWRKY18), rice (OsWRKY76), and *Solanum lycopersicum* (SlWRKY72) (Figure 2A). Multiple sequence alignments showed that ZjWRKY18 shared the highest sequence consistency with GaWRKY1 (30.4%), followed by AtWRKY40 and AtWRKY18 (26%). We constructed ZjWRKY18-GFP fusion proteins and transiently transformed them into transgenic tobacco leaves. The subcellular localization of the GFP-tagged protein was observed after 48 h or 60 h in the nucleus (Figure 2B). 

### 2.3. Suppression of ZjWRKY18 Expression Negatively Regulates Triterpenoid Pathway Genes and Reduces Triterpenoid Accumulation

To explore the role of *ZjWRKY18* in the regulation of triterpenoid biosynthesis, VIGS treatment of *ZjWRKY18* was performed in jujube fruits. RT-qPCR analysis showed that the expression of *ZjWRKY18* decreased by about 65% compared with the empty vector control (Figure 3B). The effect of *ZjWRKY18* silencing on triterpenoid accumulation was determined by HPLC, and the results showed that silencing had a significant effect on total triterpenoid accumulation (Figure 3A), and the accumulation of individual compounds (corosolic acid, betulinic acid, oleanolic acid, and ursolic acid) also decreased significantly compared with the control. Corosolic acid and betulinic acid exhibited the highest decline (approximately 53% and 49%, respectively) followed by ursolic acid (approximately 38%), and oleanolic acid (31%) (Figure 3C,D). Based on the decreased triterpenoid content caused by *ZjWRKY18* silencing, we next examined the expression of genes in the triterpenoid synthesis pathway. ZjWRKY18-VIGS samples significantly downregulated several genes in the triterpenoid synthesis MVA pathway compared with the control samples, as measured by RT-qPCR analysis (Appendix A). Among the analyzed triterpenoid synthesis genes, the acetyl-CoA acetyltransferase (*ZjAACT*) transcription level displayed a reduction of about 15%, and the transcripts of HMG-Coenzyme synthase (*ZjHMGS*) and HMG-Coenzyme reductase (*ZjHMGR1*) were reduced by about 30% and 65%, respectively. For genes encoding triterpenoid pathway intermediate synthases, farnesyl diphosphate synthase (*ZjFPS*) and squalene synthase (*ZjSQS1*) were significantly reduced, by about 68% and 34%, respectively. The expression of the downstream triterpenoid pathway gene *ZjCYP450* decreased by about 20% (Figure 3E).

### 2.4. Overexpression of ZjWRKY18 Positively Regulates Triterpenoid Accumulation and Upregulates Triterpenoid Pathway Genes

To explore whether *ZjWRKY18* overexpression promotes the expression of triterpenoid biosynthetic genes in plants, leading to an increase in triterpenoid content, a transient overexpression test was conducted in jujube fruits by agroinfiltration. Next, mRNA expression analysis of *ZjWRKY18* under the control of the CaMV 35S promoter was performed for jujube fruit infiltrated by *A. tumefaciens* for 10 days. The results showed that *ZjWRKY18* transcripts increased by about 5.5-fold (Figure 4C). The increase of triterpene synthesis pathway gene expression leads to the increase of triterpenoid accumulation (Figure 4A). The total triterpenoids increased by 2.5-fold, and corosolic acid showed the highest increase (260%), followed by ursolic acid (230%), betulinic acid (220%), and oleanolic acid (180%) (Figure 4B). Expression analysis of triterpenoid biosynthetic genes showed that most genes in the MVA pathway were upregulated, and *HMGR1*, *FPS*, and *SQS1* transcripts were the most significantly increased, by about 4.5-fold, 3.2-fold, and 1.9-fold, respectively (Figure 4C).

To further verify that *ZjWRKY18* has a positive effect on plant triterpenoids, we overexpressed *ZjWRKY18* in tobacco and *Arabidopsis* and generated transgenic plants (*ZjWRKY18-OE1*, *ZjWRKY18-OE2*, *ZjWRKY18-OE3*). In the overexpressing *Arabidopsis* and tobacco lines, the expression level of OE-*ZjWRKY18* in the transgenic lines was 25-fold to 45-fold that of the control plants (Figure 4D and Appendix A). To investigate whether *ZjWRKY18* overexpression affected the transcript levels of triterpenoid pathway genes, the expression levels of *HMGR*, *FPS*, *SQS*, *SQE*, and *OSC* in the *OE-ZjWRKY18* lines were analyzed. Compared with the control, all of the selected pathway genes in the *OE-ZjWRKY18* lines were upregulated, although to different levels (Figure 4F and Appendix A). Triterpenoid accumulation was analyzed in the roots and leaves of OE-ZjWRKY18 *Arabidopsis* and tobacco plants, as well as the control plants. The results showed that the total triterpenoid content was significantly increased (about 2-fold) compared with the control plants (Figure 4E and Appendix A). Therefore, upregulation of triterpenoid pathway genes in *OE-ZjWRKY18* may positively regulate triterpenoid accumulation in these plants, similar to the results obtained in jujube fruits. 

### 2.5. ZjWRKY18 Enhances HMGR and FPS Activities by Directly Binding to Their Promoters

To confirm whether ZjWRKY18 enhances *HMGR* and *FPS* promoter activity, we performed histochemical staining and quantitative protein activity assays. ZjWRKY18 increased the expression levels of the reporter genes ZjHMGRpro:GUS and ZjFPSpro:GUS by 2.26-fold and 2.76-fold, respectively. The GUS staining degree was consistent with the GUS protein activity pattern (Figure 5A). Using the same transient transformation reporting system, ZjWRKY18 injection activated the luciferase reporter (LUC) constructs for these genes, as ZjHMGRpro:LUC and ZjFPSpro:LUC exhibited 2.6-fold and 2.43-fold increases in LUC activity, respectively (Figure 5B). These results indicate that ZjWRKY18 can improve the activities of the *HMGR* and *FPS* promoters.

Silencing and overexpressing *ZjWRKY18* significantly downregulated and upregulated the expression of HMGR and FPS, respectively. Analysis of these promoters revealed W-box cis-acting elements, suggesting that *ZjWRKY18* might activate the transcription of these genes by binding to these elements in the promoters. We found WRKY binding sites W-box core element TTGACC in the upstream promoter regions of both *HMGR* and *FPS* located at −1685 bp and −270 bp, respectively. To confirm whether ZjWRKY18 binds to the *HMGR* and *FPS* promoters a Y1H assay was conducted. The positive control was transformed with (pAbAi-p53 + pGADT7-53), the negative control was transformed with (ZjHMGR/ZjFPSpro-pAbAi + pGADT7), and ZjWRKY18 was transformed with (ZjHMGR/ZjFPSpro-pAbAi + pGADT7-ZjWRKY18). The results showed that the yeast cells transformed with the positive control and ZjWRKY18 could grow on the SD selection medium with 600 ng/mL aureobasidin A (AbA), indicating the interaction of ZjWRKY18 with the promoters of *ZjHMGR and ZjFPS* (Figure 5C).

### 2.6. Overexpression of ZjWRKY18 Positively Regulates the Expression of Stress Genes, Leading to Increased Tolerance to Salt Stress

WRKY proteins usually respond to stress in plants, so to investigate the function of *ZjWRKY18* in plant salt tolerance, transgenic *Arabidopsis* and control samples were planted in 1/2 MS or 1/2 MS medium with 100 mM NaCl. When subjected to 100 mM NaCl stress, the seed germination of the control and transgenic *Arabidopsis* plants was inhibited to varying degrees. The seed germination rate of transgenic *Arabidopsis* plants was higher than that of the control plants, which showed wilting and yellowing, while the transgenic *Arabidopsis* plants remained green (Figure 6A). We also observed more severe inhibition of root length in the control than in transgenic *Arabidopsis* (Figure 6B,C). Transgenic tobacco also showed similar salt stress tolerance (Appendix A). After 100 mM NaCl treatment, the control grew slowly with yellow leaves, while transgenic tobacco had greener leaves and more roots. The physiological characteristics of transgenic and control plants under salt stress were also studied. The chlorophyll content of the transgenic plants was relatively high (Figure 6D and Appendix A), but the MDA content was lower in the transgenic lines than that in the control lines (Figure 6E and Appendix A). Under salt stress, SOD, POD, and CAT enzyme activities were higher in the transgenic lines than those in the control lines (Figure 6F,H,I and Appendix A). To further reveal the function of *ZjWRKY18* under salt stress, we analyzed the expression levels of genes related to stress under normal and salt stress conditions. The transcript abundances of *P5CS*, *RD29*, *DREB*, and *NCED3* in *Arabidopsis* and tobacco were significantly higher in *ZjWRKY18*-OE samples than in controls (Figure 6G and Appendix A), suggesting that the increased expression in *ZjWRKY18-OE* plants was correlated with improved tolerance to stresses and explaining how *ZjWRKY18* positively regulates salt tolerance.

## 3. Discussion

In this study, we investigated the biosynthesis and transcriptional regulation of jujube triterpenoids, and determined the functional role of *ZjWRKY18* in triterpenoid biosynthesis. Plant hormones increase the expression of the biosynthesis pathways’ genes and WRKY transcription factors act to increase the content of target secondary metabolites. For example, MeJA and SA induced the expression of *FPS*, which has been shown to be involved in triterpenoid biosynthesis, increasing the accumulation of squalene and triterpenoids in birch [29]. In *Centella asiatica*, MeJA and SA induce expression of triterpene-related genes to increase triterpenoid content [4]. In addition, silenced and overexpressed *Withania somnifera CYP450* and *Ganoderma lucidum bHLH* reduced and enhanced triterpenoid accumulation, respectively [30,31]. In this study, *ZjWRKY18* was significantly upregulated under the action of MeJA and SA, and this up-regulation in response to phytohormones, and the expression of this transcription factor in jujube, may play an important role in triterpenoid biosynthesis.

To determine the involvement of *ZjWRKY18* in the biosynthesis of jujube triterpenoids, instantaneous silencing was used to down regulate *ZjWRKY18* expression. The results showed that *ZjWRKY18* inhibited triterpenoid synthesis and reduced triterpenoid accumulation in jujube. A decrease in triterpenoid accumulation was observed after the down-regulation/silencing of the triterpene pathway genes *FPS*, *SQS*, and *OSC* [32,33,34]. *ZjWRKY18* silencing resulted in the significantly downregulated expression of several triterpenoid pathway genes, with other genes not affected, suggesting that *ZjWRKY18* may be specific for regulation of triterpenoid pathway genes. In *M. truncatula*, bHLH family TFs exhibit a different mode of regulation of triterpenoid pathway genes [35]. 

Transient overexpression of ZjWRKY18 upregulated triterpenoid pathway genes such as *AACT*, *HMGR*, *FPS*, *SQS*, *SQE*, *OSC*, and *CYP450*, resulting in enhanced triterpenoid content in jujube fruits (Figure 4). *ZjWRKY18* may interact with the promoters of triterpenoid biosynthesis related genes in plants, and interference with this interaction can regulate biosynthesis. WRKY proteins bind to W-box elements [36] to regulate gene expression, and the W-box (TTGACC) may be present in the promoters of triterpenoid MVA pathway genes. The promoter regions of *HMGR* and *FPS* in jujube contain W-box elements, and Y1H, GUS, and LUC activity assays show that *ZjWRKY18* acts to increase the activities of *HMGR* and *FPS* by binding to their promoters through direct binding of these elements. Interestingly, although the promoter of *SQS* and other genes lack W-box elements, its expression was downregulated and upregulated, respectively, in response to *ZjWRKY18* silencing and overexpression, which may be indirect effects of changes in the expression of adjacent genes. *SQS* silencing [33] and *OSC* overexpression [37] led to corresponding down- and up-regulation of triterpenoid pathway genes, showing tight regulation of triterpenoid pathway genes. The expression of *ZjWRKY18* in tobacco and *Arabidopsis* not only upregulates triterpenoid pathway synthesis genes, but also increases triterpenoid accumulation (Figure 4), suggesting that *ZjWRKY18* can regulate triterpenoid synthesis in other plant species. Transcription factors can often coordinate the transcription of multiple biosynthetic pathway genes, and the synergistic upregulation of WRKYs on secondary metabolic biosynthetic genes has also been found in other plants [38]. *CrWRKY1* overexpression in *C. roseus* upregulates terpene pathway genes to promote the accumulation of terpenoids [39] and *AnWRKY* overexpression upregulates artemisinin pathway genes, leading to increased artemisinin accumulation in *A. annua* [40]. 

Overexpression of *ZjWRKY18* increased triterpenoid content and improved plant salt tolerance, indicating that triterpene biosynthesis is related to salt stress, which is consistent with previous research results [41,42]. WRKY transcription factors can respond to abiotic stress and regulate plant stress responses in both positive and negative ways [43,44]. Overexpression of *ZjWRKY18* had a positive effect on stress response genes in jujube and model plants (Figure 6), consistent with the finding that overexpression of *MtWRKY76* increased plant salt stress tolerance in *Medicago truncatula* [45]. Stress-related genes such as *P5CS*, *DREB*, *NCED3*, and *RD29* may contribute to the high transcriptional richness of *ZjWRKY18*, which leads to increased expression of stress-related genes. We hypothesized the presence of WRKY binding sites (W-box) in the promoters of these genes. Several studies have shown that *P5CS*, *RD29*, *DREB*, and *NCED3* are stress-related genes [46,47,48,49]. Previous studies have shown that overexpression of *ZmWRKY33* in *Arabidopsis thaliana* can induce the expression of *RD29*, which is involved in the stress signaling pathway, and improve the salt tolerance of plants [50]. In addition, under NaCl stress, the seed germination rates and root lengths of transgenic *ZjWRKY18* seedlings were significantly higher than those of the wild type in early development (Figure 6). These results suggest that *ZjWRKY18* TF may play a positive regulatory role in plant response to high salt stress, which is consistent with the finding that *GhWRKY34* in *Gossypium hirsutumcan* can enhance plant salt tolerance [51]. 

Overall, *ZjWRKY18* regulates the expression of triterpenoid biosynthesis MVA pathway genes. Previous work found that citrus *CiMYB42* is a transcription factor involved in the regulation of the triterpenoid pathway [52], suggesting that the triterpenoid synthesis pathway can be regulated by different types of transcription factors in different plants. Interactions of these TFs may allow greater regulation of the synthesis and accumulation of the desired target metabolites. Biosynthesis of terpenoid indole alkaloids in *Catharanthus roseus* has been similarly reported to be regulated by multiple TFs [53]. In addition, the special metabolites regulated by WRKY transcription factors may contribute to the overall adaptability of plants to stress by enhancing their tolerance to various stresses [17]. Overall, our results highlight the importance of *ZjWRKY18* in regulating triterpenoid biosynthesis in jujube, and suggest strategies to improve triterpenoid content and salt stress tolerance in plants.

## 4. Materials and Methods

### 4.1. Plant Material Culture and Collection

Wild jujube seeds were collected from the Jujube Experimental Station, Qingjian County, Yulin, Shaanxi Province, rinsed with tap water for 24 h, and sown in sterilized soil. All wild seedlings were grown at 24 °C in a climate-controlled glasshouse (light:dark, 16 h/8 h photoperiod), three-week-old seedlings with consistent growth were selected. For phytohormone treatments, the seedlings were treated with salicylic acid (SA, 1.5 mM), methyl jasmonate (MeJA, 200 mM), or water as a control after one day’s adaptation to the hydroponic conditions. Samples were harvested at 0, 12, 24, 48, and 72 h after treatments. For transient virus-induced gene silencing (VIGS) and overexpression experiments, wild jujube fruits at 90 days after pollination were used for agroinfiltration. For transgenic plants, *Arabidopsis thaliana* and tobacco were grown in a light incubator at 22–24 °C (light: dark, 16 h/8 h photoperiod). For tissue expression analysis, roots, stems, and leaves were harvested from three-week-old wild jujube seedlings, frozen in liquid nitrogen, and transferred to −80 °C until analysis. Three biological replicates and three technical replicates were performed for all experiments.

### 4.2. Phylogenetic Analysis

A phylogenetic tree was constructed based on the sequences of WRKY TFs that regulate secondary metabolisms and some WRKYs that regulate stress tolerance (Appendix A). The phylogenetic tree was constructed using the default settings of MEGA v.6 and using the neighbor join method. The bootstrap value was calculated from 1000 repetitions [54]. 

### 4.3. Subcellular Localization Analysis

The open reading frame (ORF) sequence of *ZjWRKY18*, without the stop codon, was amplified using cDNA from wild jujube seedling leaves (Appendix A). All sequence reads have also been deposited in the NCBI online database (GCA_001835795.2). To construct the ZjWRKY18-GFP fusion plasmid, the PCR product was cloned into BamHI/SalI sites of the pCambia2300 (pC2300)-GFP vector (Appendix A). Plasmids pC2300-ZjWRKY18-GFP and pC2300-GFP were introduced into tobacco leaves as described previously [55]. Scanning GFP fluorescence imaging was performed using a confocal laser-scanning microscope (TCS SP8 SR; Leica Zeiss Germany).

### 4.4. Construction of VIGS and Overexpression Vectors and Generation of Transgenic Plants

To generate the VIGS construct, a fragment encoding *ZjWRKY18* was PCR-amplified from jujube leaves cDNA using gene-specific primers (Appendix A). By cloning the PCR product into the BamHI/XhoI sites in the pTRV2 vector, we obtained construct pTRV2-ZjWRKY18 (Appendix A). To generate the overexpression construct, the *ZjWRKY18* ORF, without the stop codon, was PCR amplified from wild jujube leaves cDNA (Appendix A) and cloned into the BamHI/SalI sites of the vector under the control of the 35S promoter of cauliflower mosaic virus (CaMV) to generate the pC2300-ZjWRKY18 construct (Appendix A). The generated VIGS and overexpression constructs were introduced into *Agrobacterium tumefaciens* strain GV3101 using a freeze-thaw technique. The triterpenoid content of the silenced and overexpressed plasmid pTRV2-ZjWRKY18, pTRV2, pC2300-ZjWRKY18, and pC2300 were measured 10 days after transient transformation into wild jujube fruit. The *ZjUBQ1* and *ZjUBQ2* genes as housekeeping genes in jujube [56]. To generate transgenic plants, plasmid pC2300-ZjWRKY18 was used to generate transgenic *Arabidopsis* according to the floral-dip method [57] and transgenic tobacco was generated by the Agrobacterium-mediated leaf disc method [58]. Transformed plants were screened by PCR and qRT-PCR and the positive lines were used for analysis. The *AtActin* and *NtActin* genes as housekeeping genes for *Arabidopsis* and tobacco, respectively (Appendix A).

### 4.5. Triterpenoids Extraction and Data Analysis

Triterpenoids were extracted following Wen et al. [5], using 0.5 mg freeze-dried samples. For triterpenoid content analysis, the extracted material was dissolved in 5 mL 90% MeOH and was analyzed using high-performance liquid chromatography (HPLC), following the method of Wen et al. [5]. The triterpenoid standards were obtained from Sigma-Aldrich and were used for identification and quantification. Total triterpenoid content was determined using the vanillin-glacial acetic acid method described by Wei et al. [59]. 

### 4.6. Glucuronidase and Luciferase Activity Assays

The *ZjHMGR* and *ZjFPS* upstream promoter sequences (approximately 2000 bp) were introduced into pC0390GUS vector and pGreen II 0800-LUC to obtain fusion plasmids. The promoter sequences are shown in Appendix A. The CDS of *ZjWRKY18* was cloned into the pGreen 62-SK vector, allowing expression of the recombinant protein pSK-ZjWRKY18. The fusion plasmids were transformed into GV3101-pSoup competent cells (Weidi, Shanghai, China). The upstream promoter and TF *Agrobacterium* suspension culture were mixed and infiltrated into the back of unfolded *N. benthamiana* leaves. The tobacco leaves were harvested for glucuronidase (GUS) staining and fluorescence signal acquisition after 48 h or 60 h of infiltration. Activity analysis was performed using a luciferase and glucuronidase reporter assay kit (Beyotime, Nanjing, China), as per the manufacturer’s instructions. Three biological and technical replicates were carried out for each sample.

### 4.7. Yeast One-Hybrid (Y1H) Assay

Y1H assays were performed following the method described previously by Huang et al. [60]. To construct the bait vectors, fragments of the promoters of the target genes *ZjHMGR* and *ZjFPS* containing the core *cis*-elements were cloned into the pAbAi plasmid. The resulting plasmids were linearized and transformed into the Y1H Gold strain and selected with medium lacking Ura (SD/−Ura medium) but supplemented with 600 ng/μL aureobasidin A (AbA) (SD/−Ura/AbA medium). To construct the prey vectors, the ORF of *ZjWRKY18* was inserted into the pGADT7 plasmid and transformed into Y1H Gold cells containing pAbAi-ZjHMGR or pAbAi-ZjFPS. The empty pGADT7-P53 and pGADT7 plasmids were used as positive and negative controls, respectively.

### 4.8. Salt Stress Treatment Analysis

For salt stress treatment, transgenic plants and wild types were treated with 100 mM NaCl or water as a control and the relevant data were recorded after the phenotype appeared (one-week-old plants). All samples were immediately frozen in liquid nitrogen and transferred to −80 °C until analysis. The activities of superoxide dismutase (SOD), peroxidase (POD), catalase (CAT), and malondialdehye (MDA) were measured using assay kits (Solarbio, China). The total chlorophyll was extracted and determined from the NaCl-treated and control tobacco and *Arabidopsis thaliana*, as described by Aktas et al. [61].

### 4.9. Statistical Analyses

One-way analysis of variance (ANOVA) and Duncan’s multiple comparison test were used to analyze the differences in triterpenoid content at different stages and different tissues using the SPSS software (version 25.0; IBM Inc., Chicago, IL, USA). In all analyses, *p* < 0.05 was taken to indicate statistical significance. Charts were plotted using GraphPad Prism 8.0.2, TBtools software and OmicShare Tools online: https://www.omicshare.com/tools/ (accessed on 16 November 2022). 

## 5. Conclusions

We performed gene silencing and overexpression studies to functionally characterize the MeJA- and SA-induced *ZjWRKY18* transcription factor and its role in triterpenoid accumulation. Silencing and overexpression of *ZjWRKY18* decreased and increased triterpenoid accumulation, respectively. In addition, we found that *ZjWRKY18* promoted activity of *ZjHMGR* and *ZjFPS* by directly binding to their promoters. Overexpression of *ZjWRKY18* increased tolerance to salt stress in tobacco and *Arabidopsis thaliana*. *ZjWRKY18* plays a positive regulatory role in plant triterpenoid biosynthesis and salt stress tolerance. The findings provide new insights to improve triterpenoid content and cultivate improved jujube varieties with high quality and strong stress tolerance (Figure 7).

## Figures and Tables

**Figure 1 ijms-24-03899-f001:**
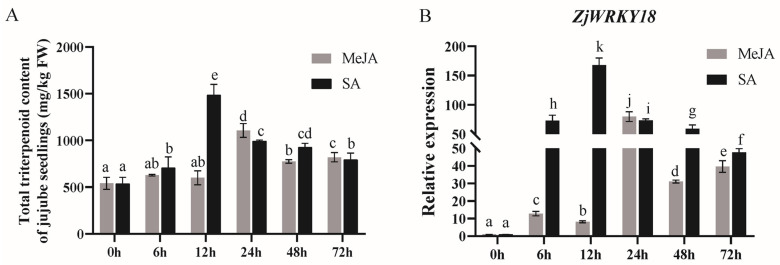
Effect of methyl jasmonate (MeJA) and salicylic acid (SA) on *ZjWRKY18* expression and triterpenoid accumulation; (**A**) HPLC quantification of total triterpenoid in wild jujube seedlings treated with MeJA and SA at different time intervals. (**B**) qRT-PCR analysis of *ZjWRKY18* expression. UBQ was used as an internal reference for normalization. The data shown are from three independent experiments. Different letters (a–k) indicate significant differences according to one-way ANOVA (*p* < 0.05). Error bars indicate mean ± SE.

**Figure 2 ijms-24-03899-f002:**
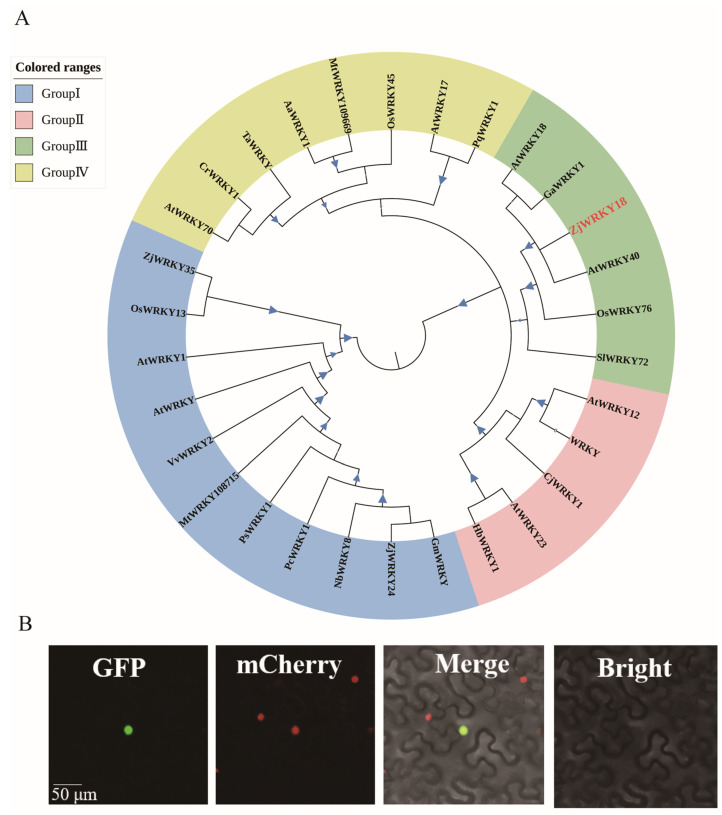
Phylogenetic tree analysis and subcellular localization. (**A**) Phylogenetic relationship of ZjWRKY18 with other plant WRKYs. (**B**) Subcellular localization of ZjWRKY18. Scale bar, 50 μm.

**Figure 3 ijms-24-03899-f003:**
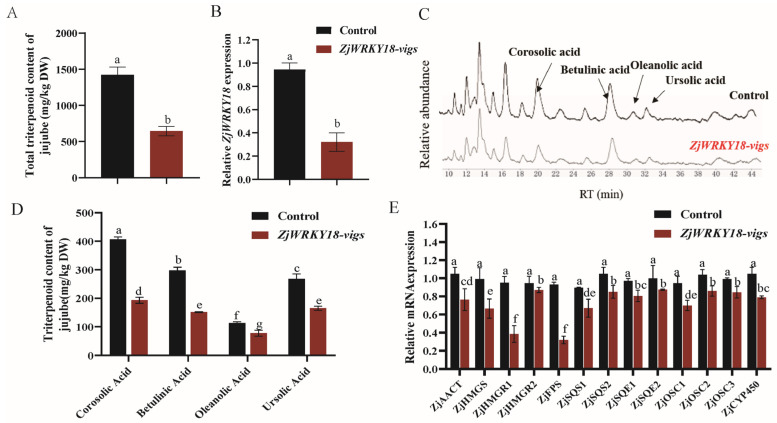
Effects of ZjWRKY18 silencing on triterpenoid content and triterpenoid synthesis pathway gene expression. (**A**) RT-qPCR analysis of ZjWRKY18 expression; (**B**,**D**) Total triterpene (**B**) and content of four triterpenoids (**D**) in control and ZjWRKY18-silenced lines. (**C**) Chromatograms showing relative triterpenoid content in ZjWRKY18-silenced lines compared with controls; (**E**) Expression levels of MVA pathway genes. Different letters (a–g) indicate significant differences according to one-way ANOVA (*p* < 0.05). Error bars indicate mean ± SE.

**Figure 4 ijms-24-03899-f004:**
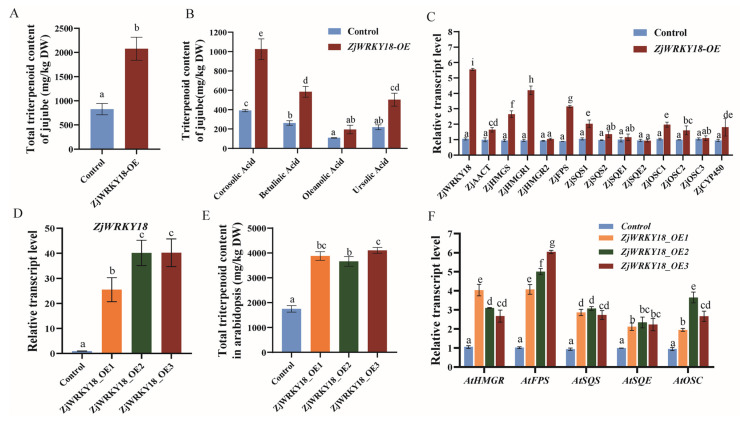
Effect of overexpression of ZjWRKY18 on triterpenoid content and the expression of MVA pathway genes; (**A**,**B**) Effect of transient overexpression of ZjWRKY18 on triterpenoid content; (**C**) Relative levels of ZjWRKY18 and mevalonate pathway genes in pC2300 control and ZjWRKY18-OE overexpression lines; (**D**,**F**) Expression of ZjWRKY18 (**D**) and triterpenoid pathway genes (**F**) in transgenic lines of *Arabidopsis thaliana*; (**E**) Total triterpene content changes in OE-ZjWRKY18 leaves compared with control. Different letters (a–i) indicate significant differences according to one-way ANOVA (*p* < 0.05). Error bars indicate mean ± SE.

**Figure 5 ijms-24-03899-f005:**
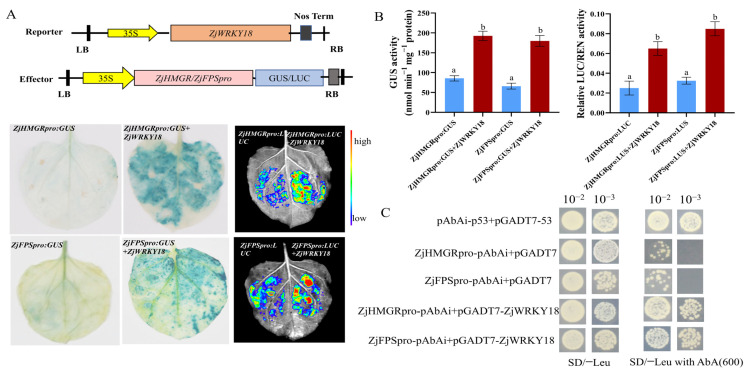
ZjWRKY18 activates the expression of ZjHMGR and ZjFPS by interaction with their promoters; (**A**,**B**) Schematic diagram of reporter and effector constructs in GUS and luciferase reporter assays. The colors represent gene promoter activities. (**C**) Y1H assay showed that ZjWRKY18 binds to the ZjHMGR and ZjFPS promoters in the yeast system. Three independent injected tobacco leaves were tested, with error bars representing SE. Different letters (a,b) indicate significant differences according to one-way ANOVA (*p* < 0.01).

**Figure 6 ijms-24-03899-f006:**
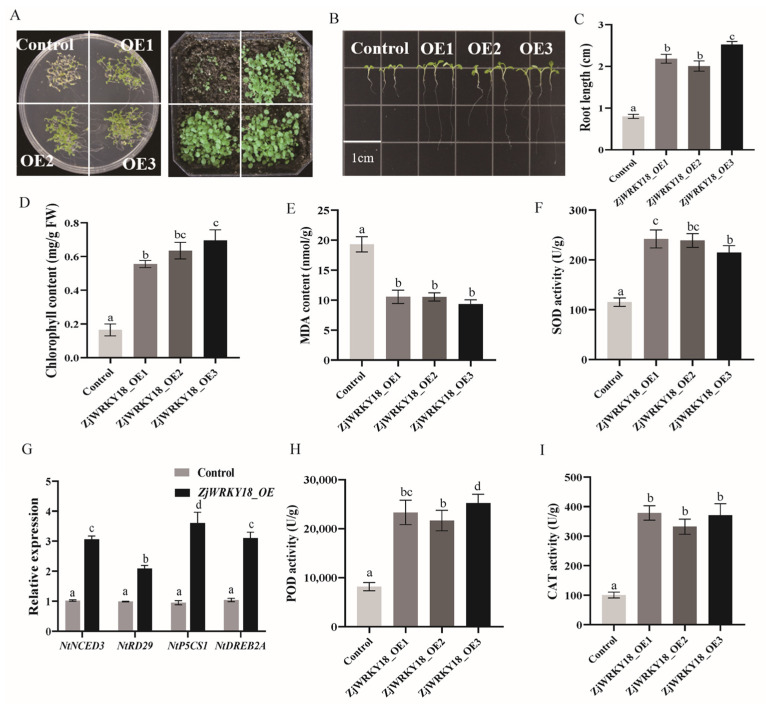
Effect of ZjWRKY18 overexpression on defense-related genes and salt stress in *Arabidopsis thaliana;* (**A**) Growth phenotypes of overexpressed ZjWRKY18 *Arabidopsis thaliana* and control under salt stress; (**B**,**C**) Root length of *Arabidopsis thaliana* overexpressed ZjWRKY18 and control under salt stress. (**D**) The difference of chlorophyll content between overexpressed and control lines; (**E**,**F**,**H**,**I**) The analysis of MDA content and SOD, POD, and CAT activities in overexpressed and control lines; (**G**) Expression levels of defense related genes. The presented data are from three independent experiments. Different letters (a–d) indicate significant differences according to one-way ANOVA (*p* < 0.05). Error bars indicate mean ± SE.

**Figure 7 ijms-24-03899-f007:**
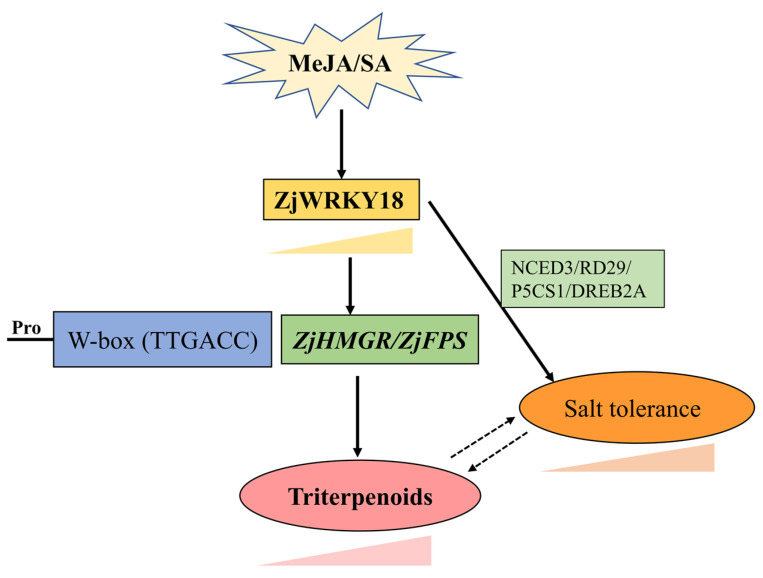
Model of *ZjWRKY18* function in triterpenoid accumulation and salt stress tolerance in jujube.

## Data Availability

Not applicable.

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
