# Peer review of "Methyl Jasmonate- and Salicylic Acid-Induced Transcription Factor ZjWRKY18 Regulates Triterpenoid Accumulation and Salt Stress Tolerance in Jujube"

_ijms, 2023, doi:10.3390/ijms24043899_

Round 1

Reviewer 1 Report

The manuscript Wen et al. observed that regulatory role of ZjWRKY18 transcription factor in triterpenoid accumulation and abiotic stress in jujube and model plants (Arabidopsis and Tobacco).

There are quires/minor corrections need to be addressed

1.    There few confusions on name control/WT OR EV plants…… Consistently use the word either control/WT throughout the manuscript.

2.    Fig. S6b (line 247) pls check the figure citation properly.

3.    Provide the primer details of salt stress responsive genes (NtNCED, NtRD29, NtP5CS1, NtDREB2A) in the table S1.

4.    Table S8. Include the details in supplementary section of the manuscript.

5.    Statisitical analysis – methods followed and details of software should be included in methods.

6. In conclusion, instead of strong stress resistance use word "stress tolerance" 

Reviewer 2 Report

Reviewer’s Comments

The submitted manuscript to IJMS entitled “Transcription factor ZjWRKY18 regulates triterpenoid accumulation and salt stress tolerance in jujube.” Although the topic of manuscript is interesting and under the scope of this journal, however, the authors made it an area-specific (Hainan) issue rather than global. Overall, based on my evaluation, I recommend the major revisions to authors.

My major concerns are;

The title should be modified and it should contain the information about MeJA and SA, too.

Figures: Please always use the significance letters rather than asterisks.

Line 432: “MeJA and SA induced ZjWRKY18 transcription factor” replace it with “MeJA- and SA-induced ZjWRKY18 transcription factor”. and please follow this scheme throughout the manuscript.

Line 434: replace “promotes” with “promoted”. Always use the past tense for your results!!!
